# ZnO ALD-Coated Microsphere-Based Sensors for Temperature Measurements

**DOI:** 10.3390/s20174689

**Published:** 2020-08-20

**Authors:** Paulina Listewnik, Mikhael Bechelany, Jacek B. Jasinski, Małgorzata Szczerska

**Affiliations:** 1Department of Metrology and Optoelectronics, Faculty of Electronics, Telecommunications and Informatics, Gdańsk University of Technology, 11/12 Narutowicza Street, 80-233 Gdańsk, Poland; malszcze@pg.edu.pl; 2Institut Européen des Membranes, IEM—UMR 5635, University of Montpellier, ENSCM, CNRS, 34095 Montpellier, France; mikhael.bechelany@umontpellier.fr; 3Conn Center for Renewable Energy Research, University of Louisville, Louisville, KY 40292, USA; jacek.jasinski@louisville.edu

**Keywords:** atomic layer deposition, fiber-optic, microsphere, temperature, ZnO

## Abstract

In this paper, the application of a microsphere-based fiber-optic sensor with a 200 nm zinc oxide (ZnO) coating, deposited by the Atomic Layer Deposition (ALD) method, for temperature measurements between 100 and 300 °C, is presented. The main advantage of integrating a fiber-optic microsphere with a sensing device is the possibility of monitoring the integrity of the sensor head in real-time, which allows for higher accuracy during measurements. The study has demonstrated that ZnO ALD-coated microsphere-based sensors can be successfully used for temperature measurements. The sensitivity of the tested device was found to be 103.5 nW/°C when the sensor was coupled with a light source of 1300 nm central wavelength. The measured coefficient R^2^ of the sensor head was over 0.99, indicating a good fit of the theoretical linear model to the measured experimental data.

## 1. Introduction

Temperature is one of the most important parameters, measured in many different fields, such as science, medicine or industry [1,2,3,4,5]. It is used to monitor quality of the products, procedures and energy consumption. Accurate temperature measurements are highly dependent on carefully chosen instruments, which should be selected based on conditions in which the device will be used, external influences and the parameters suitable for each task, especially temperature range, pollution, sensitivity and period of time, over which the measurements will be performed [6,7].

One of the fields, where the temperature is strictly controlled is the food industry. This is to ensure that the proper standards are preserved and to minimize the risks, such as bacteria growth and formation of toxins, while processing and storing the food [8,9]. Maintaining the right temperature helps to avoid food poisoning or its spoiling [10,11]. Another area, where those measurements are also highly utilized are electrochemical batteries and energy storage cells, where temperature control is used to monitor device performance and stability during charging cycles, therefore preserving its properties for as long as possible and estimating its lifetime [12,13]. Monitoring temperature is also useful during plastic or metal production to ensure the quality of the products and workplace security [14,15,16].

There is an abundance of instruments for temperature measurements, from contact sensors, such as thermocouples and thermistors to contactless devices, i.e., infrared thermal detectors [17,18,19]. Among them, all fiber-optic sensors stand out and, as a consequence, their development has been steadily progressing throughout the years thanks to the ease of use of these devices, their high durability, low fabrication cost, and chemical inertness [20,21,22]. Over the last decade, researchers gained new measurement techniques using fiber-optic sensors, such as: hybrid fiber-optic sensing, GOD-complex-based (glucose oxidase complex) sensors, multi-parameter sensors, deformable micro-mirror sensors or core-offset splicing [23,24,25,26,27,28]. The requirement for the long-term monitoring of temperature in severe, remote conditions caused fiber-optic sensors application in many fields, including industry, e.g., building applications, oil leakage railway infrastructure [29,30,31,32,33], but also they are used in biochemistry [34]. Moreover, fiber-optic sensors can be adapted to best suit the needs of specific applications, by modifying geometrical parameters or by adding additional passive components, such as coatings [35,36,37].

In traditional fiber-optic sensors, many desired properties have been hampered by low sensitivity, limited measurement range and the lack of protection against mechanical damage. In order to address these shortcomings, the sensors with various coatings (metal, metal oxides, diamond, etc.), deposited on the surface of the sensor head by different methods (atomic layer deposition, magnetron sputtering, electron beam evaporation e-beam) [38,39,40], have started to be developed in recent years.

Geometrical modifications have also been introduced to increase resolution, employ phenomena such as resonance or Whispering Mode Gallery, and control the optical path of the light within the utilized medium [41]. The most used optical-fiber structures include tapers and microspheres [42,43].

One of the most challenging aspects in remote sensing, especially under volatile conditions, is determining whether the integrity of the sensor head remains preserved. Ensuring that the sensor head keeps its integrity, it helps to eliminate inaccuracies from the obtained data and prevent major disruptions of the measurement process. Incorporating a microsphere to the fiber-optic sensor allows one to monitor the state of the sensor head during real-time measurements.

This work presents the advantages of combining a fiber-optic microsphere and ZnO (zinc oxide) ALD (Atomic Layer Deposition) 200 nm coating into one sensor, designed for temperature measurements at the central wavelength of 1300 nm.

## 2. Materials and Methods

### 2.1. Microsphere Development

The fiber-optic microsphere was manufactured at the end of a standard single-mode optical fiber (SMF-28, Thorlabs Inc., Newton, NJ, USA) by using an electric arc from the splicer (FSU975, Ericsson, Sweden), which provided sufficient energy to affect the original structure of the fiber and allowed for the microsphere to be formed by a three-step pull. During the fabrication process, the splicing parameters were carefully controlled, ensuring the high reproducibility of the microsphere structure. The diameter of the microsphere used for this study was 245 µm.

The microsphere was coated with a 200-nm-thick ZnO layer using Atomic Layer Deposition (ALD), as described elsewhere [38,44].

The described device worked as an interferometric fiber-optic sensor with an intrinsic fixed cavity. The principle of the operation of the sensor is shown in Figure 1.

During operation, part of the optical signal propagating through the fiber is reflected at the boundary between the core and the cladding of the microsphere, whereas the rest passes through and reflects off the microsphere surface. Cross-section of the sensor head presented in Figure 1b provides visualization how the signal propagates through the sensor.

These two beams interfere with each other. While the reflection on the boundary between the core and the cladding is constant, the reflection from the microsphere surface depends on the optical properties of the deposited ZnO coating. By adjusting the thermal radiation around the device, the ZnO coating is influenced, which affects the intensity of the output signal.

The sensor head has been subjected to Scanning Electron Microscopy (SEM) imaging to provide the characterization of the ZnO coating after its deposition on the surface of the microsphere. The image presented in Figure 2 is of 5000× magnification and it shows the uniformity of the ALD ZnO coating.

### 2.2. Experimental Setup

To validate the sensing capabilities of the device, experimental measurements were performed. Test measurements were performed in order to obtain a spectral response of the signal during temperature changes. The temperature was measured using a low-coherence light source, temperature calibrator and an optical signal analyzer in a configuration presented in Figure 3.

During measurements, an optical signal, that was generated and provided by the low-coherent light source—superluminescent diode with a center wavelength of 1310 nm (SLD-1310-18-W, FiberLabs Inc., Fujimino, Japan), propagated through a typical 2:1 50/50% optical coupler (G657A, CELLCO, Kobylanka, Poland) to the microsphere-based fiber-optic sensor, placed in the temperature calibrator (ETC-400A, Ametek, Berwyn, PA, USA).

The reflected signal intensity was measured in a temperature range from 100 to 300 °C with a 10 °C step. The temperature was adjusted every 5 min in order to allow the sensor to adapt to the change. After the signal was reflected by the microsphere, it was received and analyzed by the Optical Signal Analyzer (OSA, Ando AQ6319, Yokohama, Japan).

## 3. Results and Discussion

The results presented in this section were obtained according to the procedures described in Section 2.

Several series of measurements were carried out in order to test the device temperature sensing abilities. During measurements, the sensor head was placed inside the temperature calibrator, while the range was increased with 10 °C steps, between 100 and 300 °C.

Figure 4 shows an example of the measured response of the microsphere-based fiber-optic sensor at a wavelength of 1300 nm. As shown, the intensity of the reflected signal increased with the increase in the temperature. Not all of the measured responses were plotted to maintain clarity of the graph. A slight shift in wavelength (±2 nm) is a result of an optical coupler loss.

The temperature dependence of the reflected signal peak intensity measured in the entire temperature range is presented in Figure 5. Additionally, Figure 5 shows a theoretical linear fit indicating the accuracy of the device. As shown, the intensity of the reflected signal increased linearly with the temperature. The obtained coefficient R^2^, which represents the quality of the fit, is 0.995, i.e., close to 1, suggesting good agreement of the experimental data and theoretical linear fit.

The sensitivity of the sensor was calculated according to Formula (1):(1)$$S=\frac{\Delta I}{\Delta T}$$
where: S—sensitivity, Δ*I*—intensity, Δ*T*—temperature.

The sensitivity of the sensor calculated from the data in Figure 5 equals 103.5 nW/°C.

The presented results indicate that the microsphere-based sensor with 200 nm ZnO ALD coating is a promising device for temperature measurements. The described device maintains stable conditions to perform such measurements. In addition, its design provides an opportunity for constant, real-time monitoring of the integrity of the sensor head structure.

## 4. Conclusions

This paper introduces a ZnO ALD coated microsphere-based sensor for temperature measurements. The presented ZnO ALD coated microsphere-based sensor is demonstrated to be a precise device ideal for long-term monitoring temperature, which is crucial for industrial applications, such as manufacturing, processing, storing and controlling products in various sectors. Investigated sensor is a reliable device during harsh conditions and remote or hard-to-get places, where the cost of measurement system is balanced by safety considerations or service and operation costs. The presented sensor can, for example, be utilized for the investigation of processes occurring inside of electrochemical battery cells during their charging-discharging cycles. The sensor is fabricated at the end-face of an optical fiber and the coating of 200 nm in thickness is deposited on its surface by the ALD method. By using a microsphere sensor head, not only can the measured parameters be controlled, but also the structural integrity of the sensor. To optimize the metrological parameters of the device, such as sensitivity or resolution, the thickness of the coating can be modified as needed [44].

## Figures and Tables

**Figure 1 sensors-20-04689-f001:**
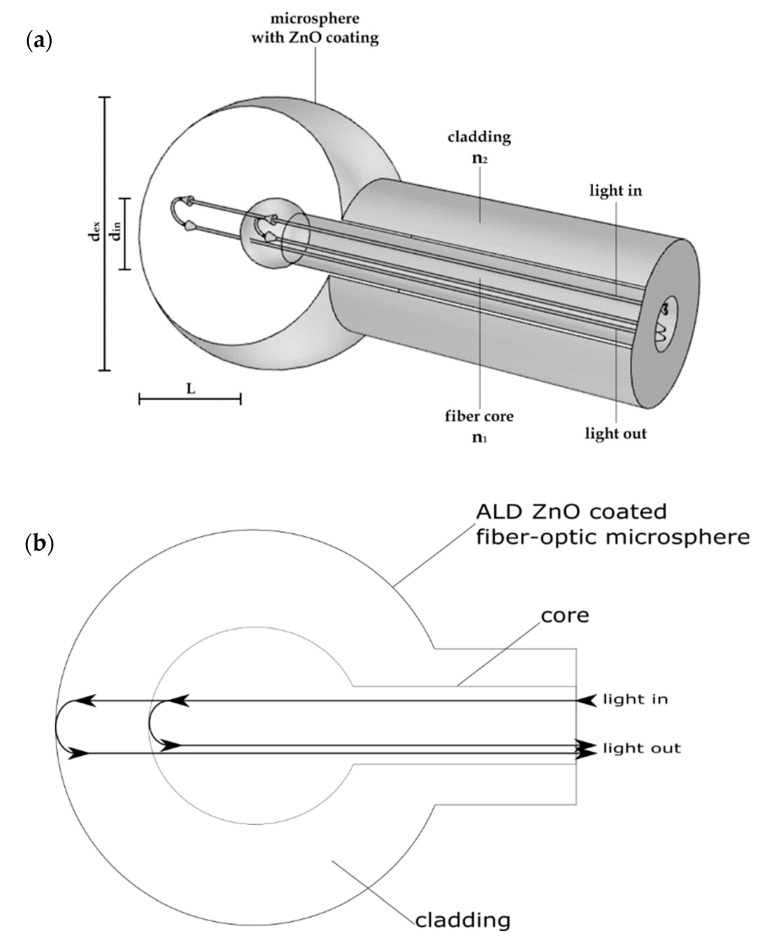
Principle of operation of the microsphere-based sensor with ZnO Atomic Layer Deposition (ALD) coating for temperature measurement: (**a**) schematic representation of a sensor, (**b**) cross-section of the sensor head.

**Figure 2 sensors-20-04689-f002:**
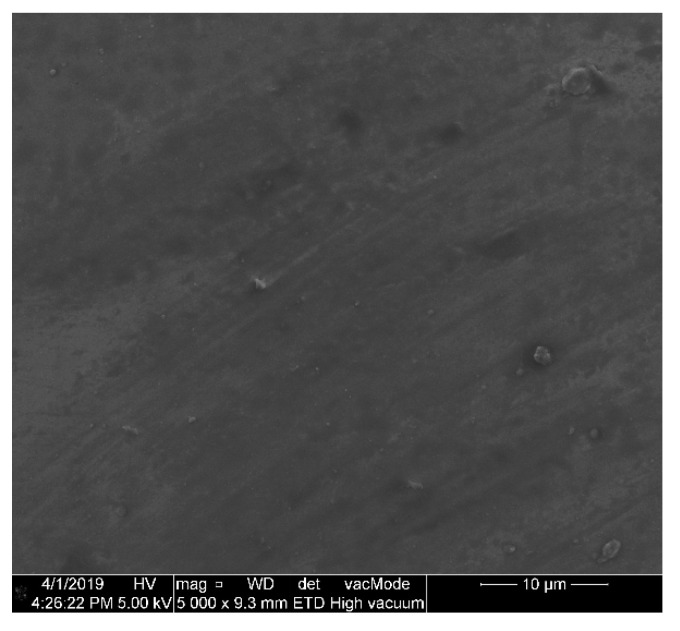
Scanning Electron Microscopy (SEM) image of the ALD ZnO coating deposited on the surface of the microsphere with 5000× magnification.

**Figure 3 sensors-20-04689-f003:**
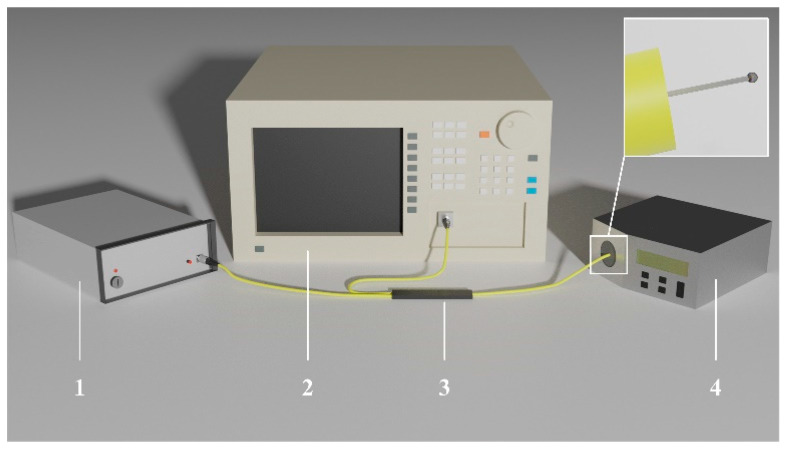
Schematic of the experimental setup used for temperature measurement, where: 1—superluminescent diode, 2—Optical Signal Analyzer, 3—optical coupler, 4—temperature calibrator.

**Figure 4 sensors-20-04689-f004:**
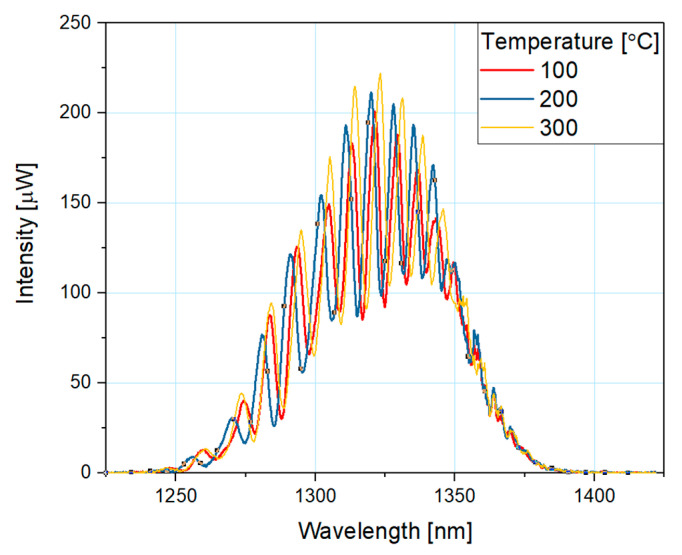
Measured response of the reflected signal intensity for the microsphere-based sensor with 200 nm ZnO ALD coating at 100, 200 and 300 °C.

**Figure 5 sensors-20-04689-f005:**
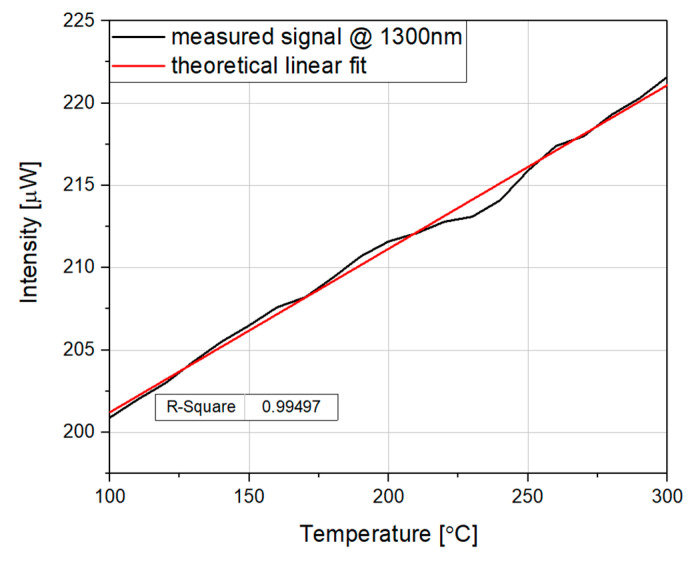
Dependence of the reflected signal peak intensity vs. temperature for a microsphere-based sensor with 200 nm ZnO ALD coating, measured at a wavelength of 1300 nm. A theoretical linear fit is also included.

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
