# Peer review of "ZnO ALD-Coated Microsphere-Based Sensors for Temperature Measurements"

_sensors, 2020, doi:10.3390/s20174689_

Round 1

Reviewer 1 Report

The authors developed a temperature sensor proposing an exciting idea about to use a sphere at the end of a SMF fiber to be used as an interferometer. The intensity dependence of the temperature Is close fit to linear with excellent sensitivity. Nevertheless, it would be desirable for the authors:

  1. Include a SEM picture of the sensor.
  2. Can provide some physical explanation about how the sensor works.

Reviewer 2 Report

The manuscript presents a temperature sensor based on a ZnO ALD-coated microsphere. The structure is new, however, the advantages of the proposed sensor over other reported fiber-optic temperature sensors is not obvious, there are concerns that should be addressed:

  1. The reflection spectrum in fig.3 indicated a multi-beam interference, the demodulation method and principle should be detailed for readers. Does the multiple frequency components affect the demodulation?
  2. The temperature test was performed in a temperature range of 100-300℃, can the sensor be employed in the environment with much higher temperature? What’s the temperature limit of the proposed sensor? The authors should give a comment on this issue.
  3. The authors said the main advantage of the proposed sensor is the possibility of monitoring the integrity of the sensor head in real-time, and the proposed sensor allows for higher accuracy during measurements. Can they discuss this in detail? And what about the accuracy?
  4. Does the thickness of coating affect the sensor performance? What’s the advantages of the proposed sensor over other fiber-optic temperature sensors that are also compact, integrated and with high temperature sensitivity?

Reviewer 3 Report

The manuscript with the title “ZnO ALD-coated microsphere-based sensors for temperature measurements” presents a novel technique for temperature measurement based on the microsphere-based fiber-optic sensor. Prior to the final decision the authors should response to the followings:

  1. The paper structure is not acceptable in the presented form since there is no clear theoretical explanation of how the sensor is working and what the advantages of such a composed sensing system are. The authors should provide more information about the basic physical principles that lie behind such a sensing system and why the microsphere is used at the fiber tip.
  2. There is no clear connection between the Figures 3, and 4., i.e. what algorithm is used for extracting the intensity shown in Figure 4. from Figure. 3. Figure 3. represents the channeled spectrum that is obtained at the fiber end where I suppose we have a low-finesse Fabry-Perot interferometer but there is no explanation where it is formed.
  3. There should be presented quantitative values of the sensor accuracy (precision). In line 121 and 122 the authors mentioned the theoretical model but this is not shown in the paper.
  4. References should be consistent since Ref. 10 and Refs. 13-18 do not have the names of the authors.

Reviewer 4 Report

Topic of the paper is of interest for the novel design of the fiber optic sensing element.

However, the contents of the proposed paper are significantly too poorly detailed.

Temperature measurements with fiber optic based sensors is widely diffused since 10-15 years, but no detailed presentation of the state of the art is provided in the introduction.

Also the "Materials and methods" section is too concise and inadequate.

No technical information is provided about the metrological performance of the employed set-up, nor an uncertainty analisys has been developed to quantify basic metrological performance of the proposed temperature sensor (measurement uncertainty, repeatability, stability...) that could help the reader to compare the proposed sensor to other optical fiber sensores navaiable for temperature measurement.

Finally, some considerations could be reported in the conclusions about the real potentials of the proposed sensor for industrial applications given the costs of the other devices in the measurement chain necessary to extract the measurement information that is coded in the output optical signal.

Reviewer 5 Report

The basis of this work is interesting and worthily to follow, but the data analysis is not convincing me.

It is not clear in the text what is the major effect in the sensor. The power reflected in the ZnO layer that changes? Or is the length/index of sphere changing?

As described, the reflected power is a result of a low finesse Fabry-Perot spectrum superposed on the SLD spectrum. The FP is very sensitive to changes in the cavity, and the fringes will move through the SLD spectrum, due to thermal change in the diameter and refractive index of the microsphere. Since there are FP peaks sweeping across the maximum emission of the SLDI would expect that a small variation of the peak power should be visible (and it is visible in Fig 4). The relative position of the fringe related to the maximum emission of the SLD (due to nanometric variation in the diameter) could influence the results (the spectra should be different for different microspheres). It is not clear for me if the measurement is made with the higher peak for each temperature, or is the peak obtained with a fitting with the fringe maximums. Another important information missing is the OSA resolution, since the peak value can change with resolution.

Sometimes it is said that the signal peak is measured other times is the intensity of the signal. This leads to some confusion. From the figures it seems that is the peak value, but the full intensity (integral of fig3) could overcome the problems with shifting spectra.

Another point that puzzles me is the FP fringe spacing. With 245 micron sphere we should expect fringes at about 2 nm spacing (is this related with line 113-114 note?) and the fringes in fig 3 show about 8 nm spacing at ~1320 nm, with a smaller spacing at higher wavelengths and larger at smaller wavelengths. This indicates that the equivalent length of the FP is around 70 microns.

Finally, along the manuscript is said several times that this design allows a control of the structural integrity of the sensor, but nothing is said how is that done. What is the difference on this for a sensor without the coating?

These are the main reasons to ask for a resubmission.

Round 2

Reviewer 2 Report

The authors have refered to most of my concerns, although some of them are not very satisfied, such as lackness of discussion on the measurement accuracy, and detailed and clearly explanation of multiple frequency components. Overall, the manuscript has been improved well and I can recommend its publication at the present form.

Author Response

Thank you for taking the time to review our paper.

Reviewer 3 Report

The manuscript is acceptable for publication in the presented form.

Author Response

(The authors gave the same response as above.)

Reviewer 4 Report

1) State of the art: An improvement in presentation of different applications of optical fiber sensors for temperatura measurements has been implemented by Authors.

Few words should be added to cite temperature measurement in biomedical applications by means of optical fiber sensors (OFS) and, also in this field, the peculiar advantages of measuring temperature by OFS as described in the following paper, useful to mention in the literature.

L. D'Acquisto, F. Scardulla, S. Pasta,
Steam sterilization processes affect the stability of clinical thermometers: Thermistor and prototypal FBG probe comparison,
Optical Fiber Technology, Volume 55, 2020, 102156

2) Although some more information about metrological qualification of the proposed  sensor could enforce the quality of the proposed communication, and consequently the interest for the reader, the proposed revision is fit for the pubblication of the proposed communucation.

Paper can be accepted after implementation of minor revision indicated at point 1) without any needs for resubmission to Reviewers.

Author Response

Thank you for taking the time to review our paper. Per Reviewer's suggestion, the paper has been cited.

Reviewer 5 Report

I'm satisfied with the revised text and with the answers to the comments.

Author Response

(The authors gave the same response as above.)
